# Effect of Irrigation and Fertilizer Management on Rice Yield and Nitrogen Loss: A Meta-Analysis

**DOI:** 10.3390/plants11131690

**Published:** 2022-06-26

**Authors:** Haonan Qiu, Shihong Yang, Zewei Jiang, Yi Xu, Xiyun Jiao

**Affiliations:** 1College of Agricultural Science and Engineering, Hohai University, Nanjing 210098, China; qhn132@hhu.edu.cn (H.Q.); zwaq@hhu.edu.cn (Z.J.); xuyi_0623@hhu.edu.cn (Y.X.); xyjiao@hhu.edu.cn (X.J.); 2State Key Laboratory of Hydrology-Water Resources and Hydraulic Engineering, Hohai University, Nanjing 210098, China

**Keywords:** irrigation schedule, nitrogen application, water–nitrogen coupling, yield, nitrogen loss

## Abstract

Irrigation and nitrogen fertilizer application are two important factors affecting yield and nitrogen loss in rice fields; however, the interaction effects of different irrigation schedules and combined management of nitrogen fertilizer application on yield and nitrogen loss in rice fields remain unknown. Therefore, we collected 327 sets of data on rice yield and 437 sets of data on nitrogen loss in rice fields from 2000 to 2021 and investigated the effects of different water-saving irrigation schedules, nitrogen application levels, and water–nitrogen couplings on rice yield, nitrogen use efficiency, and nitrogen loss (N_2_O emissions, nitrogen runoff, nitrogen leaching, and ammonia volatilization) by meta-analysis using conventional flooding irrigation and no nitrogen treatment as controls. The results showed that alternate wet and dry irrigation and controlled irrigation had increasing effects on rice yield. Alternate wet and dry irrigation had a significant yield-increasing effect (average 2.57% increase) and dry cultivation significantly reduced rice yield with an average 21.25% yield reduction. Water-saving irrigation reduces nitrogen runoff and leaching losses from rice fields but increases N_2_O emissions, and alternate wet and dry irrigation has a significant effect on increasing N_2_O emissions, with an average increase of 67.77%. Most water-saving irrigation can increase nitrogen use efficiency. Among water-saving irrigation methods, the effect of controlled irrigation on increasing nitrogen use efficiency is 1.06%. Rice yield and nitrogen use efficiency both showed a trend of increasing then decreasing with nitrogen fertilizer application, and nitrogen loss gradually increased with the amount of nitrogen fertilizer input. Water–nitrogen coupling management can significantly reduce nitrogen loss in rice fields while saving water and increasing yield. Based on the analysis of the data in this study, when the irrigation amount was 300~350 mm and the nitrogen application amount was 200~250 kg/ha, the rice yield and nitrogen fertilizer use efficiency were at a high level, which corresponded to the irrigation schedule of controlled irrigation or alternating wet and dry irrigation in the literature. However, different rice-growing areas are affected by rainfall and land capability, etc. Further optimization and correction of the adapted water and fertilizer management system for paddy fields are needed. The optimal water–nitrogen pattern of this study can achieve high rice yield and reduce nitrogen loss.

## 1. Introduction

Rice is one of the world’s major food crops, with a cultivation area reaching 13.5 × 10^9^ hm^−2^ within 87.9% of the world’s cultivated area [1,2]. High yield of rice plays an important role in ensuring world food security. Among others, irrigation and nitrogen fertilizer application are two important factors that affect rice yield [3]. Rice is a water-loving crop, and irrigation is essential for its growth. With the increasing scarcity of freshwater resources worldwide, various water-saving irrigation technologies for rice are being widely applied. Available studies have shown that various rice water-saving irrigation techniques can reduce irrigation water by 16–53.1% [4,5,6], but the effect on rice yield is variable, with alternate wet and dry irrigation increasing yield by 2.5% [7], and dry cultivation leading to a 12.3% reduction in rice yield [8]. Meanwhile, the common feature of various rice water-saving irrigation techniques is to keep the field in a water-free or thin water layer condition at a certain growth stage, and the process of alternating wet and dry will inevitably cause changes in soil physical and chemical properties compared to traditional flooded irrigated rice fields, thus affecting the process of fertilizer utilization and loss in the field. Water-saving irrigation can reduce ammonia volatilization in paddy fields, and controlled irrigation can reduce ammonia volatilization in paddy fields by 18.57% [9] but also increase N_2_O emissions. Alternating wet and dry irrigation can make N_2_O emissions increase by 79–825% [10].

In addition to the large amount of water required for rice growth, nitrogen fertilizer application is one of the important measures to maintain high and stable rice yields [11]. However, there is a problem of high fertilizer application but low fertilizer utilization [12]; consequently, a large amount of nitrogen enters water bodies and the atmosphere [13,14]. The loss of nitrogen fertilizer by leaching and runoff from agricultural fields in China is about 1.74 million tons per year [15]. N_2_O, CH_4_, and NH_3_ can affect the global climate and pollute the air [16,17]. In addition, NO_2_ and CH_4_ can also be produced when using biotechnology to treat wastewater [18], and the amount of nitrogen emitted as N_2_O through chemical nitrogen fertilizers is up to 3.35 million t per year globally [19]. The total ammonia volatilization can reach 9–40% of the total nitrogen applied to rice fields [20].

The single factors, irrigation or nitrogen application management, and part of the nitrogen losses have been investigated in many existing studies [10,12,14]. Meanwhile, water and nitrogen fertilizer are two integrated factors that can influence rice growth, development, nitrogen utilization, and translocation processes [21]. Reasonable water–nitrogen coupling can significantly improve rice yield and nitrogen use efficiency. However, the interaction effects of water–nitrogen combinations on yield and nitrogen losses remain unknown. Therefore, this study collects and organizes relevant literature data and systematically investigates the effects of different water-saving irrigation schedules, nitrogen application levels, and water–nitrogen couplings on rice yield, nitrogen use efficiency, and nitrogen loss (N_2_O emission, nitrogen runoff, nitrogen leakage, and ammonia volatilization) by meta-analysis method and gives suggestions for the optimal water–nitrogen management mode in rice fields. The results of the research can provide theoretical and technical support for the sustainable use of soil and water resources and agricultural environmental protection.

## 2. Results

### 2.1. Effect of Water and Fertilizer Management on Rice Yield and Nitrogen Use Efficiency of Paddy Field

#### 2.1.1. Rice Yield

In the study on the effect of irrigation schedules on rice yield, there were four different irrigation schedules with yield-increasing effects, including alternate wet and dry irrigation, controlled irrigation, shallow irrigation and deep storage, and shallow alternate wet and dry in Figure 1a (Appendix A). The yield-increasing effect of alternate wet and dry irrigation was significant with a mean value of 2.57%, while the yield-increasing effect of controlled irrigation, shallow irrigation and deep storage, and shallow alternate wet and dry was not significant. The irrigation schedules with yield-reducing effects were dry cultivation and moist irrigation, and the yield-reducing effect of dry cultivation was significant, reaching 21.25%, while the yield-reducing effect of moist irrigation was not significant. The effect values of all five irrigation schedules, except dry cultivation on yield, did not exceed 10%, indicating that the effect of these five irrigation schedules on yield was not significant. For nitrogen application levels that involve N2–N8 in Figure 1b (Appendix A), all seven nitrogen application levels had yield-increasing effects, among which N6 and N7 had insignificant yield-increasing effects, and the remaining several nitrogen application levels had significant yield-increasing effects, which may be due to the insufficient sample size of N6 and N7. N5 had 200–250 kg/ha of applied nitrogen. It is noteworthy that the effect of rice yield increase becomes more significant as the applied nitrogen increases, but when the applied nitrogen reaches a certain level (N5), the effect of yield increase will start to decrease. Compared with the effect of different irrigation schedules on rice yield, the effect of nitrogen application on yield is more significant.

Seven water and nitrogen combinations are shown in Figure 2 (Appendix A); all seven water and nitrogen combinations resulted in an increasing yield of rice. The yield-increasing effect of alternating wet and dry irrigation coupled with four different nitrogen application levels (W1N2, W1N3, W1N4, and W1N5) was significant, and W1N5 had the largest value of 54.81%. The yield increase effect was not significant for two water–nitrogen combinations, W2N2 and W4N4, and among the water–nitrogen combinations, W4N4 had the lowest average yield-increasing effect of 11.56%.

#### 2.1.2. Nitrogen Use Efficiency

Since the nitrogen use efficiency was calculated using the no nitrogen application treatment as the control, this section no longer uses ln*RR* as the effect size but instead the *d* (Hedges’ d) in the MetaWin software as the effect size.

Three different irrigation schedules were included in this part, namely alternate wet and dry, controlled irrigation, and dry cultivation (Figure 3a, Appendix A). Alternate wet and dry and controlled irrigation can improve nitrogen use efficiency, while dry cultivation had a decreasing effect on nitrogen use efficiency with a mean value of 1.64%, but it is non-significant. Similarly, all six nitrogen application levels (N2~N6, N9) improve nitrogen use efficiency (Figure 3b, Appendix A), among which the average effect of N9 on improving nitrogen use efficiency was the largest, reaching 15.35%. It is worth noting that, except for N9, the relationship between yield and nitrogen use efficiency under nitrogen applications was similar.

The significant water–nitrogen combinations in improving nitrogen use efficiency were all the combinations shown in Figure 4 (Appendix A). The largest average effect was W1N5, which reached 16.26%, indicating that the effect of water–nitrogen coupling on improving nitrogen use efficiency was more significant than that of the two single factors of the irrigation schedule and nitrogen application rate.

### 2.2. Effect of Water and Fertilizer Management on Nitrogen Loss

#### 2.2.1. N_2_O Emission

Alternate wet and dry, controlled irrigation, shallow irrigation and deep storage, and shallow alternate wet and dry were involved in this part, and all four irrigation schedules in Figure 5a (Appendix A) increased N_2_O emissions from paddy fields. The irrigation schedules that had significant effects on N_2_O emissions were alternate wet and dry and shallow alternate wet and dry, while controlled irrigation and shallow irrigation and deep storage did not significantly affect N_2_O emissions from rice fields. There were three levels of nitrogen application, N2, N3, and N4, and these three levels of nitrogen application had a significant effect on the increase of N_2_O emissions (Figure 5b, Appendix A). Significantly, the N_2_O emissions increased with the increase of nitrogen application; N4, i.e., 150–200 kg/ha, had an N_2_O increase effect of 92.86%.

From Figure 6 (Appendix A), where we can see the five water–nitrogen combinations, it can be observed that the effect of the five water–nitrogen combinations on increasing N_2_O emissions is significant, among which W2N2 has the smallest N_2_O emissions, its average effect value being 76.03%. The effect of water–nitrogen coupling on N_2_O emissions is more significant.

#### 2.2.2. Nitrogen Runoff Loss

There were four irrigation schedules, namely alternate wet and dry, controlled irrigation, moist irrigation, and shallow irrigation and deep storage. It can be seen from Figure 7a (Appendix A) that alternate wet and dry, controlled irrigation, and moist irrigation reduced nitrogen runoff losses, and the effects of alternate wet and dry and moist irrigation were significant, while shallow irrigation and deep storage increased nitrogen runoff losses but not significantly. It is noteworthy that the effect of moist irrigation on reducing nitrogen runoff losses reached 66.21%, while the effect of shallow irrigation and deep storage on increasing nitrogen runoff loss reached 52.13%. For nitrogen applications, N2–N7 were involved in this part (Figure 7b, Appendix A). The amount of nitrogen runoff loss was caused by all six levels of nitrogen applications, among which the increasing effect of N3, N4, and N5 on the amount of nitrogen runoff loss was significant, and the increasing effect of N5 on the amount of nitrogen runoff loss was 2.94 times higher than no nitrogen application. Compared to the effect of the irrigation schedules on nitrogen runoff loss, the increasing effect of all six nitrogen application levels on nitrogen runoff loss in this study exceeded 100%, indicating that the effect of nitrogen application on nitrogen runoff loss was more significant than that of irrigation schedules.

Two water–nitrogen combinations were included in the study of nitrogen runoff loss, and both nitrogen application levels (W1N3 and W1N5) under alternating wet and dry irrigation increased nitrogen runoff loss from rice fields, but the effect of both water–nitrogen combinations on increasing nitrogen runoff loss from rice fields was not significant, and the water–nitrogen combination with the least runoff loss was W1N5, whose average effect value was 2.82 times higher than W0N0 (Appendix A).

#### 2.2.3. Nitrogen Leaching Loss

From Figure 8a (Appendix A) we can see that alternate wet and dry, controlled irrigation, shallow irrigation and deep storage, and thin and wet irrigation were involved in this part. All four irrigation schedules reduced nitrogen leaching from the rice field. Alternate wet and dry, controlled irrigation, and thin and wet irrigation had significant effects, and the effect of controlled irrigation on the reduction of nitrogen leaching reached 49.13%. The reduction effect of shallow irrigation and deep storage on nitrogen leaching from paddy fields was not significant. There were four levels of nitrogen application, N3, N4, N5, and N7, and nitrogen leaching was increased by all four levels of nitrogen application (Figure 8b, Appendix A). N4 and N5 increased nitrogen leaching significantly. For the effect of increasing nitrogen leaching, N4 was 1.08 times higher than N0. N3 and N7 showed a non-significant increase in nitrogen leaching. This phenomenon may be due to the lack of abundance of sample points. In addition, comparing different irrigation schedules, water-saving irrigation could reduce nitrogen leaching losses, while different levels of nitrogen application caused an increase in nitrogen leaching losses.

W1N3 and W1N5 were included, and both water–nitrogen combinations were the same as in the nitrogen runoff study; both water–nitrogen combinations increased nitrogen leaching losses but were not significant, and W1N3 had the smallest nitrogen leaching, its average effect value being 6.91% (Appendix A). Nitrogen leaching loss also showed a superimposed effect of two single factors compared to the effect of two single factors.

#### 2.2.4. Ammonia Volatilization

Figure 9a (Appendix A) shows the effect of irrigation schedules on ammonia volatilization. There were two irrigation schedules, alternate wet and dry and controlled irrigation, and ammonia volatilization was reduced by these two irrigation schedules. The effect of controlled irrigation on the reduction of ammonia volatilization was significant, and the effect value of controlled irrigation on the reduction of ammonia volatilization was significantly higher than that of alternate wet and dry, with an effect value of 20.97%, while the effect value of alternate wet and dry on the reduction of ammonia volatilization was only 0.44%. For nitrogen application, N3–N7 were involved in this part (Figure 9b, Appendix A), an increase in ammonia volatilization was caused by all five levels of nitrogen applications, and the increasing effect of ammonia volatilization was significant. It is worth noting that the increasing effect of ammonia volatilization was 27.49 times higher than N0 when the nitrogen application level was N7, i.e., 300–350 kg/ha. Compared with the effect of irrigation schedule, it can be seen that the application of N is the main factor that causes a significant increase in ammonia volatilization.

Three water–nitrogen combinations were included, all three water–nitrogen combinations were under alternate wet and dry irrigation at different levels of nitrogen application, i.e., W1N3, W1N4, and W1N5, and all three water–nitrogen combinations resulted in increased ammonia volatilization, but none of the effects of increased ammonia volatilization were significant, which may be influenced by the sample size. The water–nitrogen combination with the smallest effect of increased ammonia volatilization was W1N4, with an average effect value of 2.14 times higher than W0N0 (Appendix A).

### 2.3. Relationship between Water and Fertilizer Management on Nitrogen Loss and Yield

The relationship between water nitrogen input and rice yield and nitrogen uptake utilization was analyzed by fitting using a nonlinear surface. The fitting results are shown in Figure 10 and Figure 11, from which it can be seen that both rice yield and nitrogen fertilizer utilization showed a trend of increasing then decreasing with the increase of water and nitrogen inputs; therefore, water and nitrogen inputs promoted the growth of rice. However, excessive water and nitrogen inputs not only did not lead to further yield increase but also caused a large amount of nitrogen loss, resulting in lower nitrogen fertilizer use efficiency. When the irrigation amount is 300–350 mm and the nitrogen application amount is 200–250 kg/ha, the rice yield and nitrogen fertilizer uptake efficiency are at a high level, which corresponds to the irrigation schedule of controlled irrigation or alternating wet and dry irrigation in the literature. However, different rice-growing areas are affected by rainfall and land capability, etc., so we need to further optimize and correct the corresponding water and fertilizer management system for rice fields.

## 3. Discussion

### 3.1. Effect of Water-Saving Irrigation on Rice Yield and Nitrogen Loss in Paddy Fields

In the study of the effect of irrigation schedules on yield as well as nitrogen loss, there are a large number of studies based on alternate wet and dry irrigation and controlled irrigation, which have richer sample points. Alternate wet and dry irrigation as a water-saving irrigation method has started to replace conventional flood irrigation widely [22]. Alternating wet and dry irrigation can significantly reduce the amount of irrigation frequency as well as irrigation water compared to conventional irrigation [4,23], while for yield, alternating wet and dry irrigation will increase rice yield [22,24], though some research differed, indicating that alternating wet and dry irrigation will decrease rice yield [25,26]. Results from different test sites may vary. For example, when alternate wet and dry irrigation is carried out in northern China, it may have a yield-increasing effect of 10% versus flood irrigation, while in southern China, the effect value may be smaller or show a yield reduction. The result of this paper is a comprehensive study of the results of multiple test sites. In this paper, alternating wet and dry irrigation had a significant yield increase effect on rice with an average effect of 2.57%, and alternating wet and dry irrigation can improve nitrogen utilization in rice fields because moderate alternating wet and dry irrigation can regulate the balance of soil water and oxygen around the rice root system to a certain extent, which in turn affects the conversion of soil nitrogen [27]. This paper showed that controlled irrigation can improve rice yield compared to conventional flood irrigation, which promotes the growth and accumulation of dry matter in rice and facilitates nitrogen uptake and utilization [28].

Alternating wet and dry irrigation significantly increased N_2_O emissions, which is consistent with the findings of Sibayan E.B. et al. [29]. The reason for the greater N_2_O emissions from alternate wet and dry irrigation than conventional flooding is that there are two main types of greenhouse gases emitted from rice fields, CH_4_ and N_2_O, which show the relationship of trade-off [30], that is, the longer the duration of flooding, the greater the CH_4_ emissions and the smaller the N_2_O emissions. Conventional flood irrigation has to undergo a long period of flooding; therefore, the N_2_O emissions are hindered, while the greater the fluctuation of soil moisture content, the greater the N_2_O emissions [31]. For nitrogen loss, alternating wet and dry irrigation can significantly reduce nitrogen runoff with an average effect value of 32.79%, which is higher than that of controlled irrigation, which is consistent with the results of existing studies [32]. However, alternating wet and dry irrigation can easily cause leaching losses due to the frequent oxygen-enriched-anoxic transformation in the soil [33]. The results of this paper showed that the effect value of alternating wet and dry irrigation on reducing nitrogen leaching losses was 14.24%, which was lower than the effect value of controlled irrigation of 49.13%. The initial period (10 d) after transplanting of rice is a critical period for controlling nitrogen loss, which is often influenced by different rainfall, soil types, and crop types [34]. Therefore, the use of different water and nitrogen management measures for rice fields in different regions is important to reduce nitrogen losses from rice fields. Compared to N_2_O emissions, ammonia volatilization is an important pathway for gaseous losses of nitrogen in rice fields, and ammonia volatilization in rice fields can account for 9–40% of nitrogen application [9]. In this study, the effect of controlled irrigation on reducing ammonia emissions from rice fields was significant, with an average effect value of 20.97%, which is consistent with the findings of Peng et al. [35], and the effect of alternating wet and dry irrigation on reducing ammonia emissions was not significant. Under conventional irrigation, ammonia volatilization occurs at the interface between the water layer and the atmosphere, whereas under controlled irrigation conditions, most ammonia volatilization occurs at the soil surface. With water-saving irrigation, the water layer from presence to absence brings a lot of nitrogen into the deep soil layer [36].

### 3.2. Effect of Optimized Water and Fertilizer Management on Rice Yield and Nitrogen Loss

Nitrogen application can significantly increase rice yield, but nitrogen loss also increases with nitrogen application, and ammonia emission from paddy fields increases with nitrogen application. In this investigation, the increased effect of ammonia volatilization from paddy fields was 27.49 times higher than N0 when N7, which showed an extremely high level of increase, because when nitrogen application is high, nitrogen will mainly be in the form of NO_3_-N in all soil profiles [37]. Ammonia volatilization shows an exponential increase with the amount of nitrogen applied [20], which is generally consistent with the results of this paper. The global annual N_2_O emissions due to nitrogen fertilizer use are as high as 3.35 million t [19], and the results of this research indicate that N_2_O emissions also increase with the level of nitrogen application, which is consistent with the results of Zhao et al. [38].

The maximum yield-increasing effect of rice was observed when the N application level was N5, with an average effect value of 57.08%, which is consistent with the findings of Liang et al. [39]. This paper showed that water–nitrogen coupling significantly increased rice yield as well as nitrogen use efficiency due to enhanced photosynthesis, delayed protein decomposition in rice, and increased rice root vigor under the water–nitrogen coupling effect [40].

This paper focuses only on the effect of nitrogen application level on rice yield and nitrogen loss, while reasonable optimized fertilizer application not only improves soil environment but also increases rice’s stress tolerance [41]. Small amounts of optimized fertilizer several times can increase early rice yield by 7% and reduce nitrogen loss by 14% compared to farmers’ customary fertilizer application [42], which is because reduced nitrogen application, as well as nitrogen fertilizer transport, can reduce the soil’s nitrate-nitrogen level, thus reducing nitrate-nitrogen leakage to deeper soils [43]. Although nitrogen application several times improves fertilizer efficiency, its diffusion is limited due to increased operational and labor costs [44]. Controlled-release nitrogen fertilizer is a fertilizer that controls the rate and timing of nutrient release through different regulatory mechanisms to extend the validity of nutrient uptake and utilization by plants [45]. Controlled-release nitrogen fertilizer could increase late rice yield by 9.68–27.72% and improve nitrogen use efficiency by 27.95–31.10% compared to conventional fertilization methods [46]. In addition, the use of some organic fertilizers instead of chemical fertilizers can also increase rice yield; the use of 10% organic nitrogen instead of pure chemical fertilizers for nitrogen application under optimized fertilization conditions resulted in a two-year average yield increase of 5.1% and a 27.7% increase in fertilizer use efficiency compared to conventional fertilizers [47].

## 4. Materials and Methods

### 4.1. Data Collection

The data that covered the effects of different irrigation systems and different levels of nitrogen application on rice yield, as well as nitrogen loss, were collected from the China National Knowledge Internet (CNKI) and Web of Science for Meta-analyses. We conducted the searches with the following keywords: “paddy” + “water nitrogen ”; “paddy” + “water fertilizer”; “paddy” + “ammonia”; “paddy” + “nitrogen runoff”; “paddy” + “nitrogen seepage”; “paddy” + “N_2_O”. All the articles are from 2000 to 2021 were searched and screened for the following criteria: (1) The study must be a field trial under conventional flood irrigation + no nitrogen application versus different irrigation schedules (alternating wet and dry, shallow wet irrigation, etc.) + different nitrogen application levels. (2) The study should record at least one or more data on yield, nitrogen use efficiency, N_2_O emissions, nitrogen runoff, nitrogen leaching, and ammonia volatilization. (3) Articles with duplicate trial data should be excluded. After screening again according to this criterion, a total of 74 articles met the criteria. The data needed for this paper (i.e., yield, N_2_O emissions, etc.) were extracted from tables of the articles or using GetData Graph Digitizer software if statistical graphs such as line graphs and bar graphs were given. The resulting data were grouped according to the irrigation schedule and nitrogen application to facilitate subsequent subgroup analysis, and the details of the grouping are shown in Table 1, field moisture control standards of all the schedules are shown in Table 2.

### 4.2. Meta-Analysis Statistics

In this paper, a meta-analysis was conducted using MetaWin software, with *lnRR* (response ratio) as the effect size [48], as a response to the effect of different irrigation schedules compared to conventional irrigation on rice yield, N_2_O emissions, nitrogen runoff, nitrogen leakage, and ammonia volatilization. In the case of nitrogen application grouping, the response is the effect of different nitrogen application levels compared to no nitrogen application on yield and other data, and all equations in this section are illustrated in terms of yield.
(1)$$\ln RR=\frac{X_{t}}{X_{c}}$$

Here, *X_t_* is the mean yield of the treatment group (kg/ha), *X_c_* is the yield of the control group (kg/ha), and ln*RR* is a dimensionless unit.

The within-study variance corresponding to each study effect size was calculated as:(2)$$V_{\ln RR}=\frac{S_{t}^{2}}{N_{t}X_{t}^{2}}+\frac{S_{c}^{2}}{N_{c}X_{c}^{2}}$$
where *S_t_* is the standard deviation of the yield of the treatment group, *S_c_* is the standard deviation of the yield of the control group, *N_t_* is the number of samples of the yield of the treatment group, and *N_c_* is the number of samples of the yield of the control group; if the standard deviation is labeled in the literature, the standard deviation is used directly, and if the standard deviation is not labeled in the literature, the standard deviation is calculated by referring to the following method. If the study provides multiple years of experimental data, the data from different years of the trial are considered as replicate trials; otherwise, the standard deviation is assumed to be 1/10 of the mean.

Considering that the data came from different locations with various hydrological, and rice differences, this paper used a random model in MetaWin software for analysis to obtain the mean effect value ln*RR++*. Studies with less than three sample points in the subsequent forest plotting were excluded.

After that, according to the calculated 95% confidence interval, if the 95% confidence interval contains 0, the difference between the treatment and control groups is not significant, and if it does not contain 0, the difference is significant.

For ease of understanding, the following equation is utilized:(3)$$Z=(e^{\ln RR}-1)\times 100$$

The response ratios were converted into percentages. Taking the irrigation schedule as an example, if the *Z* value is positive, it means that the irrigation schedule has a yield-increasing effect compared to conventional irrigation. If the *Z* value is negative, it means that the irrigation schedule has a yield-decreasing effect compared to conventional irrigation. If the *Z* value is closer to 0, it means that the irrigation schedule has less effect on yield.

For the calculation of nitrogen use efficiency, the *d* (Hedges’ d) was used as the effect size, using the following equation:(4)$$d=\frac{\left(X_{t}-X_{c}\right)}{S}J$$
(5)$$J=1-\frac{3}{4(N_{t}+N_{c}-2)-1}$$
where *X_t_* is the nitrogen use efficiency of the treatment group (%), *X_c_* is the nitrogen use efficiency of the control group (%), *S* is the sample standard deviation, calculated by MetaWin software, and *J* is the correction factor.

The within-study variance corresponding to each study effect size was calculated as:(6)$$V_{d}=\frac{N_{c}+N_{t}}{N_{c}N_{t}}+\frac{d^{2}}{2(N_{c}+N_{t})}$$
where *N_t_* is the number of nitrogen use efficiency samples in the treatment group and *N_c_* is the number of nitrogen use efficiency samples in the control group. The mean effect value *d++* was obtained using MetaWin software, where a positive *d* value indicates that the treatment group has an increasing effect on nitrogen use efficiency compared to the control group; otherwise, a negative *d* value indicates that the treatment group has a decreasing effect, and if the *d* value is closer to 0, the smaller the effect of the treatment on nitrogen use efficiency.

The data analyses in this paper were all performed in MetaWin and plotted using Origin software.

## 5. Conclusions

(1)All water-saving irrigations except dry cultivation and moist irrigation were able to improve rice yield in this research. The yield increase effect of alternating wet and dry irrigation was significant with an effect value of 2.57%, while the yield reduction effect of dry cultivation was significant with an effect value of 21.25%. Nitrogen application can more significantly affect the yield level. The yield will increase with nitrogen application, and the maximum effect of yield increase is 57.08% when N5 is reached. Water–nitrogen coupling increases yield, and the results showed that the greatest effect of yield increase was observed for W1N5, with an effect value of 54.81%.(2)Some water-saving irrigation can increase nitrogen use efficiency. Both alternating wet and dry and controlled irrigation significantly increased nitrogen utilization with an effect value of 0.47% and 1.06%, respectively, while dry cultivation decreased nitrogen use efficiency with an effect value of 1.64%. The effect of nitrogen application on nitrogen use efficiency was the same as that on yield. N9 has the highest nitrogen use efficiency with an effect value of 15.35%. Furthermore, the highest water–nitrogen combination for improving nitrogen use efficiency was W1N5, with an effect value of 16.26%.(3)Part of nitrogen loss can be reduced by water-saving irrigation. Alternate wet and dry irrigation reduced nitrogen runoff and leaching losses from paddy fields with effect values of 32.79% and 14.24% but increased N_2_O emissions from paddy fields with an effect value of 67.77%, while controlled irrigation also reduced nitrogen runoff and leaching losses from paddy fields with effect values of 19.31% and 49.13% and reduced ammonia volatilization with an effect value of 20.97%. Different levels of nitrogen application all caused increased nitrogen loss. W2N2 had the lowest N_2_O emissions with an average effect value of 76.03%. W1N5 had the lowest nitrogen runoff losses with an average effect value of 2.82 times higher than W0N0, W1N3 had the lowest nitrogen leaching losses with an average effect value of 6.91%, and W1N4 had the lowest ammonia volatilization with an average effect value of 2.14 times compare to W0N0.(4)This research finds an optimal water–nitrogen pattern that when the irrigation amount was 300–350 mm and the nitrogen application amount was 200–250 kg/ha, the rice yield and nitrogen use efficiency were at a high level, and the corresponding irrigation schedule in the literature was controlled irrigation or alternating wet and dry irrigation. However, the corresponding irrigation schedule still needs to be further modified to suit the rice fields in different areas, considering the effects of different regions and rainfall.

## Figures and Tables

**Figure 1 plants-11-01690-f001:**
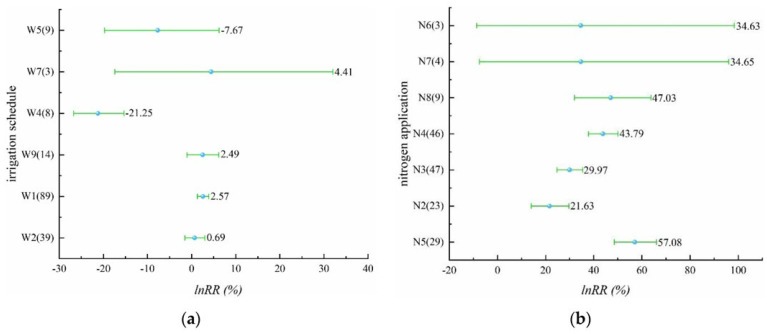
(**a**) Effect of different irrigation schedules on rice yield (*p* = 0.00000); (**b**) effect of different nitrogen application levels on rice yield (*p* = 0.00011).

**Figure 2 plants-11-01690-f002:**
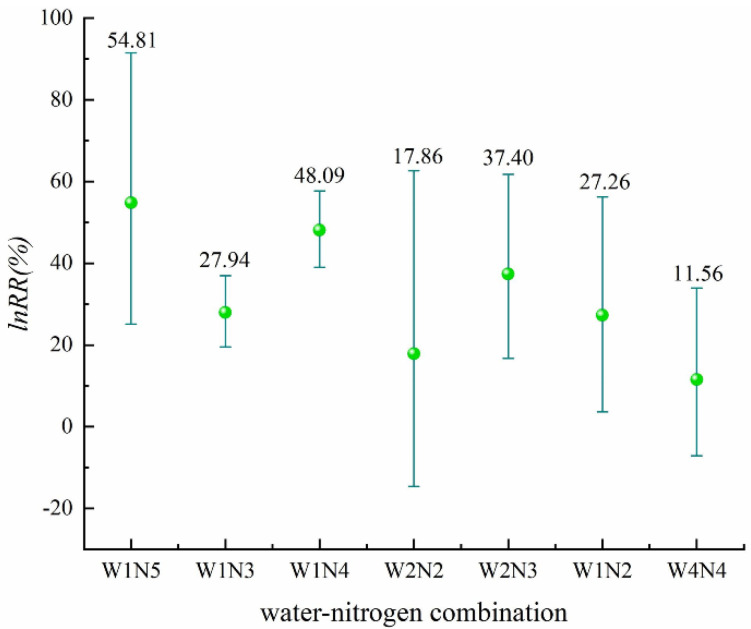
Effect of different water–nitrogen combinations on rice yield (*p* = 0.00987).

**Figure 3 plants-11-01690-f003:**
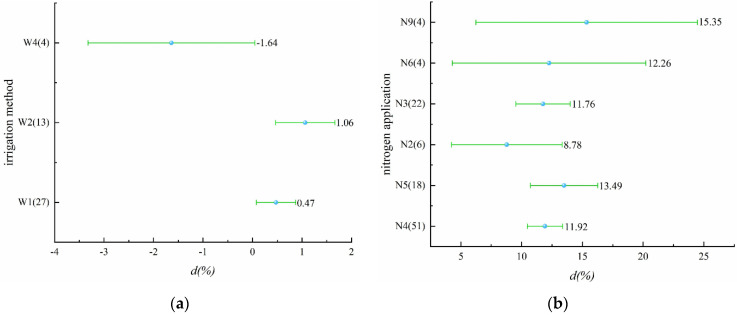
(**a**) Effect of different irrigation schedules on nitrogen use efficiency (*p* = 0.01682); (**b**) effect of different nitrogen application levels on nitrogen use efficiency (*p* = 0.03588).

**Figure 4 plants-11-01690-f004:**
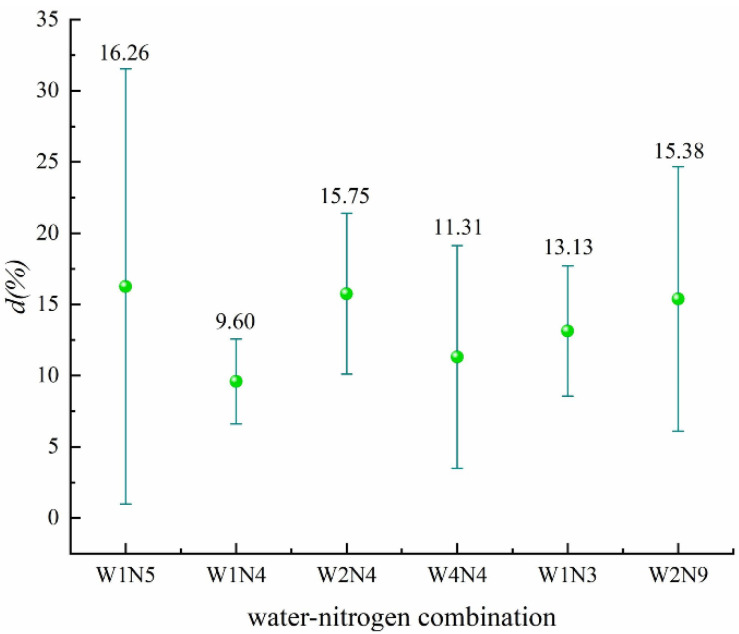
Effect of different water–nitrogen combinations on nitrogen use efficiency (*p* = 0.01631).

**Figure 5 plants-11-01690-f005:**
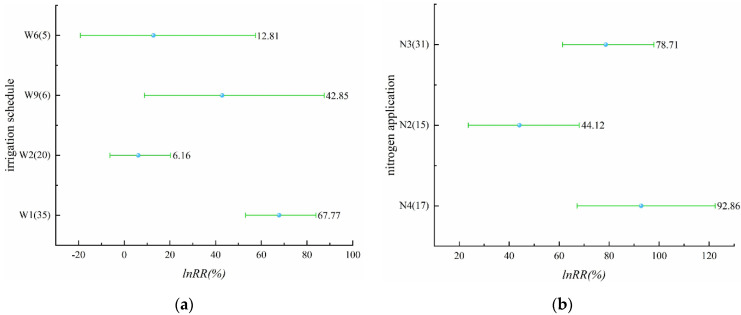
(**a**) Effect of different irrigation schedules on N_2_O emissions (*p* = 0.00004); (**b**) effect of different nitrogen application levels on N_2_O emissions (*p* = 0.00538).

**Figure 6 plants-11-01690-f006:**
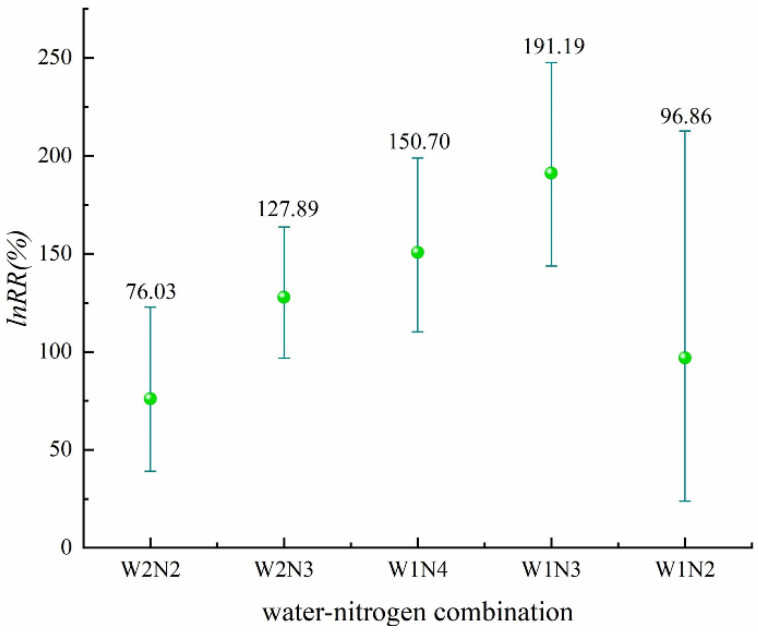
Effect of different water–nitrogen combinations on N_2_O emissions (*p* = 0.00000).

**Figure 7 plants-11-01690-f007:**
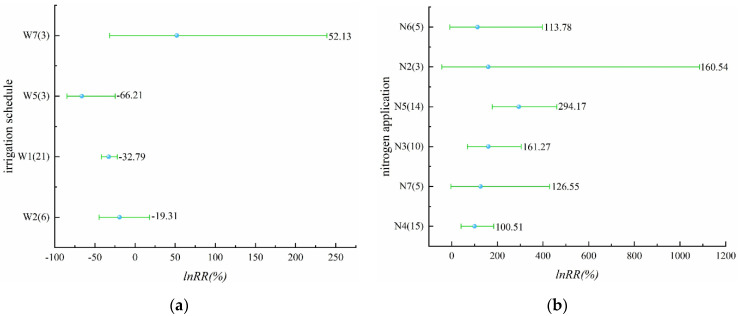
(**a**) Effect of different irrigation schedules on nitrogen runoff (*p* = 0.00138); (**b**) effect of different nitrogen application levels on nitrogen runoff (*p* = 0.52096).

**Figure 8 plants-11-01690-f008:**
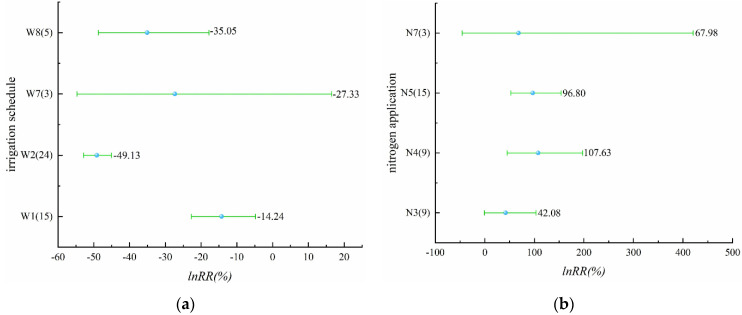
(**a**) Effect of different irrigation schedules on nitrogen leaching (*p* = 0.00000); (**b**) effect of different nitrogen application levels on nitrogen leaching (*p* = 0.44466).

**Figure 9 plants-11-01690-f009:**
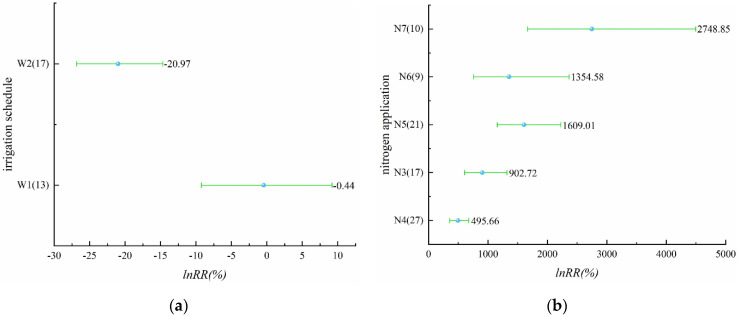
(**a**) Effect of different irrigation schedules on ammonia volatilization (*p* = 0.02987); (**b**) effect of different nitrogen application levels on ammonia volatilization (*p* = 0.00000).

**Figure 10 plants-11-01690-f010:**
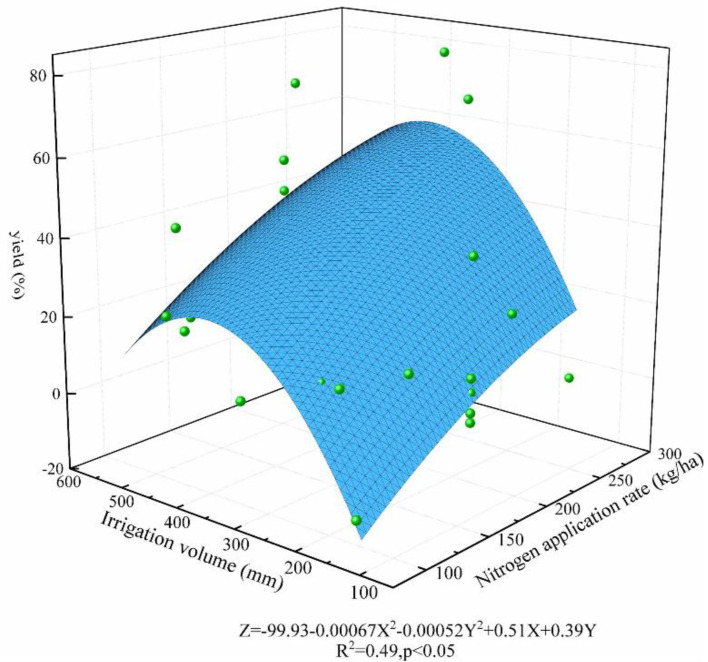
Effect of water–nitrogen coupling on yield.

**Figure 11 plants-11-01690-f011:**
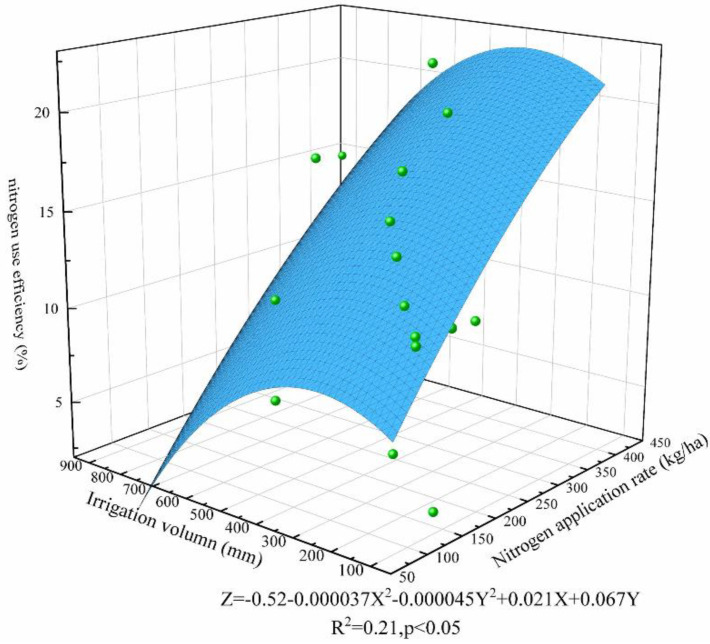
Effect of water–nitrogen coupling on nitrogen use efficiency.

**Table 1 plants-11-01690-t001:** Data grouping status.

Irrigation Schedule	Nitrogen Application Rate (kg/ha)
Flood irrigation (CK)	N0 (0)
Alternate wet and dry (W1)	N1 (0~50)
Controlled irrigation (W2)	N2 (50~100)
Mild alternate wet and dry (W3)	N3 (100~150)
Dry cultivation (W4)	N4 (150~200)
Moist irrigation (W5)	N5 (200~250)
Drip irrigation (W6)	N6 (250~300)
Shallow irrigation and deep storage (W7)	N7 (300~350)
Thin and wet irrigation (W8)	N8 (350~400)
Shallow alternate wet and dry (W9)	N9 (400~450)
-	N10 (450~500)

The irrigation schedules in Table 1 are all the irrigation schedules involved in the valid data collected.

**Table 2 plants-11-01690-t002:** Field moisture control standards at the growth period under different irrigation schedules.

Mode	Con	G ^a^	T_I_	T_M_	T_L_	J/B	H/F	M	R
CK	L ^b^	0–20	0–20	0	dry	0–30	0–10	0–10	dry
	U	20–30	20–50	50	dry	20–60	20–50	20–50	dry
W1	L	0–20	dry2-5d	0	dry	dry2-5d	dry2-5d	dry2-5d	dry
	U	20–30	20–30	10	dry	20–60	20–60	20–60	dry
W2	L	5–10	70%θs	65%θs	60%θs	75%θs	80%θs	70%θs	dry
	U	25–30	θs ^c^	θs	θs	θs	θs	θs	dry
W3	m^3^/ha	0	340			327	351	342	0
W5	Irrigate to 2 cm each time when no water layers, timely drainage on rainy days.								
W6	650–700 m^3^/667 m^2^ for the whole growth stage and stopped 20 d before harvest.								
W7	50 for storage		Irrigate 40–60 after drying to 100 below the topsoil				150 for storage when rain		
	when rain		100 for storage when rain						
W8	L	5–10	0.8θs	\	0.7θs	0.9θs	0	0.8θs	dry
	U	30	20	\	20	30	30	20	dry
W9	L	20	0.75θs	\	0.60θs	10	5	0.70θs	dry
	U	30	10	\	0	20	15	10	0

^a^ G, T_I_, T_M_, T_L_, J/B, H/F, M, and R represent regreening stage, initial tillering stage, middle tillering stage, late tillering stage, jointing and booting stage, heading and flowering stage, milk stage, and ripening stage, respectively. ^b^ Where L-limit and U-limit represent the lower limit of irrigation and the upper limit of irrigation. If not specified, the corresponding unit is mm. ^c^ θs indicates the saturated water content of the soil.

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
