# Peer review of "Effect of Irrigation and Fertilizer Management on Rice Yield and Nitrogen Loss: A Meta-Analysis"

_plants, 2022, doi:10.3390/plants11131690_

Round 1

Reviewer 1 Report

please find attached file for my comments.

Reviewer 2 Report

Current work states irrigation and nitrogen fertilizer application as two important factors affecting yield and  nitrogen loss in rice fields. Alternate wet and dry irrigation had a significant yield increasing effect (average 2.57% increase) and dry cultivation significantly reduced rice yield with an average 21.25% yield reduction. – Give standard deviation of your results, mean, max and min.

"yield-reducing effect of W4 was significant, reaching 21.25%, while the yield-reducing effect of W5 was not significant- give p- values

Better introduction to Results is needed using W1-W7 in more meaningflul way depicting their meanings and results better. Water-saving irrigation reduces nitrogen runoff and leaching losses from rice fields but increases N2O emissions. This paper presents the latest developments on the N fertilization system based on energy efficient processes using new insights into N biotransformation involving metabolic pathways, N2O, its elimination and functional microorganisms involved during the formation of process.

 Figures quality is bad and they could cut their amount. Sharpen it up, too. Error bars on Fig. 2 seem very large, to see any significant difference.

„Among them, The significant effects“ -caption not needed within sentence

Fig. 4- no units on y axis.

„kg·hm-2“- check the unit, is it „Ha“ or hectar better use?

„dry irrigation can make N2O emissions increase by 79%-825% [9]“- maybe 82.5%?

 After „There are also greenhouse gases such as N2O and CH4 as well as NH3 into the atmosphere, which has an impact on the global climate as“ some works can be cited: Literature has shown different environmental treatments to be solved for more economic way, which could be shown: DOI: 10.1016/j.biortech.2016.02.051, https://doi.org/10.3390/w13030350, https://doi.org/10.1080/09593330.2013.874492

Check caption and spacing mistakes such as: „irrigation will increase rice yield[22,23] „ and elsewhere

Table 2 hard to understand? Less words need to be used in W4-W7.

P values and standard deviations should be included in Your MS.

"seven water and nitrogen combinations were shown"- sentence starts with capital.
